# *Physalis peruwiana* Fruits and Their Food Products as New Important Components of Functional Foods

**DOI:** 10.3390/ijms26083493

**Published:** 2025-04-08

**Authors:** Beata Olas

**Affiliations:** Faculty of Biology and Environmental Protection, Department of General Biochemistry, University of Lodz, Pomorska 141/3, 90-236 Lodz, Poland; beata.olas@biol.uni.lodz.pl; Tel./Fax: +48-42-6354485

**Keywords:** biological activity, goldenberry, *Physalis peruviana*, pro-healthy potential

## Abstract

*Physalis peruviana* is a native evergreen plant from the Andean region. It is also commonly known as goldenberry and gooseberry in English-speaking countries. *P. peruviana* fruit is a globose berry, yellowish in color, which has a pleasant smell and taste. In addition, fruits of this plant have been identified as a priority part for commercialization (also for their food products: wine, jam, and juice). The health benefits of *P. peruviana* are related to the content of various bioactive chemical compounds, including withanolides, phenolic compounds (especially flavonoids), alkaloids, sucrose ester, and others such as vitamins, especially carotenoids, and physalins. The aim of the present mini-review is to provide an overview of the beneficial potential of *P. peruviana* fruits and their food products, especially fruit juice, as important components of functional foods.

## 1. Introduction

*Physalis peruviana* is a native evergreen plant from the Andean region. It is also commonly known as goldenberry and gooseberry in English-speaking countries. It belongs to the Solanaceae family. *P. peruviana* fruit is a globose berry, yellowish in color. It is about two inches in diameter and its juicy pulp, which has a pleasant smell and taste, is enclosed in a five-sepal calyx. In addition, fruits of this plant have been identified as a priority part for commercialization (also for their food products: wine, jam, and juice). Fruits have a relatively long shelf life if stored properly; at 8 °C, it can be as long as 62 days [1].

Mokhtar et al. [2] indicated the following composition of *P. peruviana* powder waste: 16.7% dietary fiber, 61% carbohydrates, 15.9% protein, 13.7% fat, 5.9% moisture, and 3.5% ash. The fatty acid profile demonstrated that linoleic acid was the main fatty acid. Moreover, high levels of tyrosine/phenylalanine, histidine, and cysteine/methionine were noted.

In addition, *P. peruviana* fruits were a good source of vitamin C compared to other fruits (orange, mango, and guava), but vitamin C content decreased after bleaching. For example, vitamin C was assessed by high-performance liquid chromatography (HPLC) method by Vaillant et al. [3]. The vitamin E content was also high, especially in *P. peruviana* fruit juice. More details about the nutritional values of *P. peruviana* are described by Kasali et al. [4].

Around 502 phytoconstituents were identified in various parts of *P. peruviana*, especially its fruit (38.2%) extracts. Several chemical classes were found in *P. peruviana*, including especially esters (11.5%), phenolic compounds (15.0%), and terpenes (26.1%). However, the pro-health benefits of this plant are related to the content of various bioactive chemical compounds, including 76 withanolides, 5 phenolic compounds—flavonoids, 2 alkaloids, 14 sucrose ester, and others, such as vitamins, especially carotenoids, and physalins, which were identified by chromatographic methods [5]. In this plant, withanolides (which are a class of polyoxygenated steroids based on C_28_ ergostane skeleton, and are characterized by the presence of a δ- or γ-lactone ring formed by a carboxylic group at C-26 and hydroxyl groups at C-23 or C-22) are prominent components, and they have been isolated from its aerial parts, whole plant, fruits, calyces, and roots [5,6,7,8,9]. Recently, Cong et al. [10] have isolated from the whole plants of *P. peruviana* a new withanolide—withaperuvin O. It is important that withanolides are chemical compounds, which have demonstrated various bioactivities, in particular the inhibition of tumors [11]. The review paper of Kasali et al. [8] demonstrates more details about the chemical compounds identified from different parts of *P. peruviana*.

Standard techniques have been used for phytochemical analysis of *P. peruviana* fruits or other plant parts, including HPLC with ultraviolet (UV) detection, liquid chromatography–tandem mass spectrometry (LC-MS/MS, gas chromatography–mass spectrometry (GC-MS), and liquid chromatography–mass spectrometry (LC-MS) [3]. The selected bioactive compounds of *P. peruviana* fruits, seeds, and pomace are presented in more detail in Figure 1.

Various parts of this plant, including twigs, leaves, stem, fruits, flowers, roots, seeds, and others are often used in traditional medicine in different countries (for example, Cameroon, Colombia, India, Kenya, Nepal, Tanzania, Uganda, and others), but leaves are the mostly used part (49.3%), prepared by decoction (31.6%) and overall administrated orally (53.6%) as the main route of admission [7,8,12,13]. Uses of various parts of *P. peruviana* in ethnomedicine are summarized in Figure 2. For example, they have antispasmodic, antidiabetic, diuretic, antiseptic, sedative, and analgesic properties. In addition, they are also used to treat hepatitis, malaria, dermatitis, asthma, and rheumatism. Recently, a review paper by Althubyani and Alrefai [14] described the protective effects of various plants, including *P. peruviana*, against food additive-induced toxicity.

Few review papers have explored the ethnotherapeutic uses of various parts of *P. peruviana* [7,8], but these papers generally do not include *P. peruviana* fruits and their food products as important components of functional foods. For example, a review by Nocetti et al. [7] only describes the biological activities of selected goldenberry byproducts, including seeds and pomace, which are produced after juice extraction. The aim of the present mini-review is to provide an overview of the beneficial potential of *P. peruviana* fruits and their food products as important components of functional foods.

## 2. Research Methods

Different databases such as SCOPUS, ScienceDirect, Web of Science, Web of Knowledge, PubMed, and Sci Finder were searched for papers examining bioactive ingredients with the beneficial potential of. *P. peruviana* fruits and their food products. The following search terms were used: “*P. peruviana*”, “goldenberry”, “gooseberry”, “fruit”, “juice”, “seed”, “pomace”, or a combination of the terms. Full articles in the English language were retrieved without time limit restriction. Recent papers were evaluated first. The last search was run on 1 April 2025.

### 2.1. Bioactive Compounds of Fruits and Their Biological Activity

The fruits contain an appreciable amount of lipids and carotenoids (especially β-carotene), with some tocopherols (about 17 mg/100 g of fruit FW, mainly ɤ-, α-, and β-tocopherol), phytosterols (about 10 mg/100 g of fruit FW, mainly campesterol and 5-avenasterol), and lutein diesters [15,16,17].

Various withanolides and triterpenes were not only isolated from leaves and roots, but also from *P. peruviana* fruits, but at low concentration [5,18]. In addition, the extract from *P. peruviana* fruits contained various flavonoids (including kaempferol, quercetin, rutin, and myricetin), sucrose esters (peruvioses and nicoandrose), and other chemical compounds such as blumenol A, hydroxyester disaccharides, carbohydrate ester of cinnamic acid, glycosically bound compounds, (+)-(*S*)-dehydrovomifoliol, and loliolide with a different biological activity [5]. Moreover, *P. peruviana* fruits are the source of trans-resveratrol, which is even richer than red wine [19].

### 2.2. Antioxidant Activity (In Vitro and In Vivo)

Navarro-Hoyos et al. [13] studied the phytochemical profile of *P. peruviana* fruits from Costa Rica, using ultra-performance liquid chromatography coupled with high-resolution mass spectrometry (QTOF-ESI MS). They identified 66 chemical compounds, including 9 flavonoids, 23 sucrose ester derivatives, and 34 withanolides. In addition, they found the role of phenolic compounds in antioxidant activity of *P. peruviana* fruits, using the DPPH method. In this experiment, dry plant material was extracted by a mixture of ethanol–ethyl acetate (75:25) as a solvent (at 125 °C).

Horn et al. [20] observed the antioxidant action of an aqueous extract from *P. peruviana* fruits in erythrocytes exposed to acid 2,4-dichlorophenoxyacetic in vitro. For example, this extract (1 and 10 g/L) decreased the lipid peroxidation measured by the level of thiobarbituric acid reactive substances (TBARS).

Hassan et al. [21] observed the antioxidant activity and free radical-scavenging of *P. peruviana* fruit juice (1 mL/kg bw/day) in a hepatocellular carcinoma rat model. The used juice had positive effects, such as reduced oxidative stress as well as improvement in the cellular antioxidant defense system and antioxidant status. According to Dewi et al. [22], *P. peruviana* fruit juice (5 and 25 mL/kg/day) improved antioxidant and adiponectin levels of high fat diet in rats (n = 6) treated with strepozotocin. However, the juice (25 mL/kg/day) presented better effects than the juice (5 mL/kg/day), but quercetin (30 mg/kg/day) showed the best antioxidant effect. Other authors also noted antioxidant potential of extracts from *P. peruviana* fruit in various in vitro and in vivo models [23,24], but there are no concrete clinical studies. Therefore, it may be a hot topic in the future, especially for investigating the antioxidant potential of *P. peruviana* fruit juice.

### 2.3. Antidiabetic Activity (In Vitro and In Vivo)

Pino-de la Fuente et al. [25] observed that the administration of *P. peruviana* pulp extract (with a dose of 300 mg/kg bw/day, for 8 weeks) improves insulin resistance in skeletal muscles by reducing both serum insulin and sugar concentrations in blood from obese mice (induced by a regular diet rich in fats (60% fat, 20% protein, 20% carbohydrate)). This extract also improved the inflammatory state and protected the liver against oxidative stress. In this experiment, *P. peruviana* fruits were pressed with peel and seeds and homogenized in a blender to obtain pulp. Also, a hydroalcoholic extract from *P. peruviana* fruits and its various fractions (ethyl acetate, hexane, and ethyl acetate residue) demonstrated antidiabetic properties in streptozotocin-induced diabetic rats after 28 days of supplementation. In addition, supplementation with this extract reduced high levels of alkaline phosphatase, alanine aminotransferase, aspartate transaminase, glycosylated hemoglobin, and glucose [26]. Rey et al. [27] also noted inhibitory effects of an extract from *P. peruviana* fruits on some intestinal carbohydrates (for example, α-amylase (IC_50_—619.9 g/mL) and α-glucosidase (IC_50_—4191 µg/mL)). A clinical trial of Rodriguez Ulloa and Rodriguez Ulloa [28] showed that the supplementation of *P. peruviana* juice increases glucose clearance in young adults.

Angel-Martin et al. [29] also observed that a regular consumption of *P. peruviana* fruits effectively prevents insulin resistance and obesity in hyperglycemic rats (n = 64). The consumption of these fruits (5% (*w*/*w*)) was conducted using fresh fruits, which were carefully chopped into pieces to facilitate their intake by the animals. The tested diet resulted not only in blood glucose concentrations, but also normalized plasma biochemical profiles (including HDL, LDL, triglyceride, and cholesterol). Moreover, this diet modulated specific urinary parameters (especially pipe colic acid—a primary marker for type 2 diabetes). In addition, authors also noted the impact of *P. peruviana* fruits on crucial metabolic regulatory genes, including lipoprotein lipase (LPL), peroxisome proliferator-activated receptor gamma (PPAR-ɤ), fatty acid synthase (FASN), and insulin receptor (INSR).

Recently, Gonzalez-Buenrostro et al. [30] studied the effect of saline stress (10–40 mM) on the metabolic profile and antidiabetic activity of *P. peruviana* fruits. They observed that saline stress (especially 40 mM NaCl) increases the phenolic content in *P. peruviana* fruits (for example, about 39% increase for the 40 mM NaCl treatment). On the other hand, fruits (treated with saline stress during cultivation) did not significantly decrease the hyperglycaemic peak in healthy rats.

However, most studies investigating antidiabetic properties lack the necessary analysis of individual chemical compounds. Vaillant et al. [3] studied metabolites after acute and chronic intake of *P. peruviana* fruits in healthy adults (at baseline, at 6 h after the intake of 250 g of *P. peruviana* fruits (acute intervention), and after 19 days of consumption of 150 g of *P. peruviana* fruits/day (medium-term intervention). They identified 49 and 36 discriminant metabolites after the acute and medium-term interventions, respectively. In addition, they suggest that *P. peruviana* fruits intake may be associated with insulin signaling (mainly involving insulin, epidermal growth factor receptor (EGFR), and the phosphatidylinositol 3-kinase pathway (PI3k/Akt/mTOR).

Bernal et al. [31] observed the hypoglycemic activity of chemical compounds (sucrose esters, named as peruvioses C-E, at the concentration of 640 µg/mL) isolated from *P. peruviana* fruits, using the α-amylase inhibition test. Peruviose D demonstrated the highest activity, with an inhibitory activity value of 84.8%. Other studies indicate that four other chemical compounds, rutin, chlorogenic acid, withaperuvin F and H, are alpha-amylase inhibitors from *P. peruviana*, and are promising antidiabetic candidates [32].

### 2.4. Anti-Cancer Activity (In Vitro and In Vivo)

Various authors observed the antiproliferative action of extracts from *P. peruviana* fruits against lung cancer cells (A549), Hep G2, Hep 3B, and PLC/PRF/5 human hepatoma cell lines in in vitro models [33,34,35,36]. For example, high concentrations of *P. peruviana* ethanolic extract (800 μg/mL) exhibited significant anti-cancer activity against lung (A549) cells [35]. This extract contained phenols (125.4 mg/g DW), flavonoids (6.39 mg/g DW), tannins (14.8 mg/g DW), alkaloids (3.37 g/100 g DW), anthocyanins (6.68 μg/100 g FW), and carotenoids (1.53 mg/100 g FW). In addition, this extract also exhibited antioxidant and antimicrobial activity against gram-positive and gram-negative bacteria. According to Ramadan et al. [37], the ethanolic extract from *P. peruviana* fruits was more powerful in inhibiting colon cell lines (IC_50_—142 gµ/mL) than the breast cell line (IC_50_—37 µg/mL) in vitro.

Wu et al. [33] observed apoptotic morphological changes such as chromatin condensation and DNA fragmentation in human Hep G2 cells treated with the ethanolic extract from *P. peruviana* fruits (50 µg/mL) in vitro. Moreover, authors indicated that the apoptosis induced by the tested extract is possibly mediated through multiple pathways, suggesting many compounds, rather than a single component.

The analysis of the expression of mRNA by Mier-Giraldo et al. [38] demonstrated the changes of antiapoptotic genes in the presence of two extracts from *P. peruviana* fruits (ethanol and isopropanol crude extract) in an in vitro model. The presence of ursolic acid and rosmarinic acid was found in both extracts, but gallic acid, quercetin, and epicatechin were higher in the isopropanol extract. In addition, a relationship was noticed between the total polyphenol content, antioxidant activity, and cytotoxic activity that was dependent on the solvent used. An isopropanol extract from *P. peruviana* fruits exhibited a half maximum inhibition concentration (IC_50_) value of 66.2 ± 2.7 mg/mL for murine fibroblast (L929) cells, and 60.5 ± 3.8 mg/mL for cancer cells of human cervix (HeLa). Moreover, two tested extracts showed immunomodulatory potential; they reduced the release of interleukin-6, interleukin-8, and monocyte chemo-attractant protein (MCP-1) in a dose-dependent manner.

Results of El-Kenawy et al. [34] demonstrated that the administration of an extract from *P. peruviana* fruits (150 mg/kg/day, for 16 weeks) protects against carcinogenesis induced by nicotine-derived nitrosamine ketone (a key tobacco smoke carcinogen) in rats (n = 60), but authors did not describe the chemical characteristics of the used plant extract. Serag et al. [39] observed that juice from *P. peruviana* fruits is more effective than adriamycine (the reference anti-cancer drug) in rats with hepatocellular carcinoma (n = 30). Rats were administered this juice daily (at a dose of 1 mL/kg/bw per week).

However, the compounds responsible for anti-cancer properties have not been well identified due to the considerable variation present in the chemical composition of *P. peruviana* fruit extracts, which can depend on inter alia time of harvest and extraction technique. Only a few papers have described these properties. Sayed et al. [40] suggest that magnolin (isolated from 95% ethanol extract of *P. peruviana* fruits) is a potent preferential antiproliferative compound against the human pancreatic cancer cell line PANC-1 with IC_50_ of 0.51 ± 0.46 µM (it was comparable with positive control—doxorubicin (IC_50_ of 0.17 ± 0.15 µM)). It is important that magnolin had much less cytotoxicity in comparison with doxorubicin (for the concentration 5 µg/mL: 6.96% growth inhibition (for magnolin), and 30.48% growth inhibition (for doxorubicin)) towards dermal fibroblasts (in vitro). In this experiment, MTT (3-(4, 5-dimethylthiazolyl-2)-2, 5-diphenyltetrazolium bromide) assay was used. Moreover, magnolin (25–100 nM) induced a concentration-dependent suppression of the formation of PANC-1 colonies. It also limited PANC-1 tumor cell migration. An in silico model indicates that matrix metalloproteinase-3 (MMP3) is the molecular target for magnolin. This compound inhibited the catalytic activity of MMP3 (IC_50_ of 185 ± 4.86 nanomolar). In the same context in an in vitro model, Wang et al. [36] observed that magnolin (10–100 µM) suppressed the phosphorylation of MEK1/2 (MAP/ERK kinase), extracellular signal-regulated kinase ½, significantly downregulated the expression of cyclin-dependent kinase 1 (CDK1), the antiapoptotic B-cell lymphoma 2 (BCL2), and metastasis-associated matrix metalloproteases 2 and 9, and upregulated the cleaved caspases 3 and 9 in breast cancer cells (MDA-MB-231). Lee et al. [41] found the same signaling pathway in lung cancer cells. Another group [42] noted that magnolin (15–60 µM) also inhibits human ovarian cancer through targeting the ERK1/2 signaling pathway.

Recently, Kim et al. [43] observed that magnolin (administered intraperitoneally: 0.1, 1 or 10 mg/kg/day, for 8 days) inhibits paclitaxel (2 mg/kg/day, for 8 days)-induced cold allodynia and ERK1/2 activation in mice (n = 12).

### 2.5. Anti-Inflammatory Activity (In Vitro and In Vivo)

In the study of Moya et al. [44], the effect of an extract from *P. peruviana* fruits on the expression of inflammatory parameters in the Caco-2 cell line was evaluated. Treatment of Caco-2 cells with the used plant extract (80 µg/mL, for 72 h) induced a reduction in IL-18, MCP-1 mRNA expression, but no effects on toll-like receptor 4 (TLR4) and NF-κB mRNA expression were observed. It is interesting that the antiarthritic activity of an extract from *P. peruviana* fruits was associated with the inhibition of inflammatory mediators [45]. The anti-inflammatory properties of this extract were assessed against cyclooxygenase 1 (COX-1) and cyclooxygenase 2 (COX-2) activity in an in vitro model. This extract exhibited inhibition against COX-1 and COX-2 activities (IC_50_ of 2.90 ± 0.1 μg/mL and 1.97 ± 0.061 μg/mL, respectively, as compared to reference standards indomethacin (0.57 ± 0.1 and 0.103 ± 0.01 μg/mL), respectively). The tested extract from *P. peruviana* fruits had not only anti-inflammatory potential in vitro, but also in vivo (at concentration—1000 mg/kg/day, for 3 weeks, in rats). In addition, authors suggest that steroidal lactones withaperuvin E/C and hydroxywithanolide E are promising lead compounds for the inhibition of TNF converting enzyme (in silico study).

According to Pardo et al. [46] juice from *P. peruviana* fruits had moderate anti-inflammatory properties by inducing pterygium formation in rabbit eyes (compared to the reference compound—methylprednisolone).

Recently, Natania et al. [47] studied the potential of plant-derived exosome-like nanoparticles (PDENs) from *P. peruviana* fruit (PENC) form human dermal fibroblast regeneration and remodeling. Isolated PENC was 190–220 nm in size, and was non-toxic up to a concentration of 500 µg/mL (using MTT assay). In addition, PENC (2.5–7.5 µg/mL) induced human dermal fibroblast proliferation and migration, upregulated collagen I production, and decreased matrix metalloproteinase-1 (MMP-1). Another author [48] observed the anti-inflammatory potential of PENC (averaged 227.7 nm in size) through the ability to reduce M1 macrophages product and promote M2 polarization.

Recently, Aboulthana et al. [49] investigated the role of metal nanoparticles (ZnO-NPs) in improving the efficiency of *P. peruviana* juice (composition of vitamins: vitamin C (9.13 mg/100 g) and vitamin B_3_ (3.15 mg/100 g) have been the most abundant in this juice; 14 phenolic compounds: ferulic acid, ρ-coumaric acid, and ellagic acid—the main identified phenolic acids; catechin, quercetin 3-*O*-rhamnoside, apigenin 7-*O*-glucoside, rutin, and apigenin 7-neohespirosde—the major identified flavonoids), and they have found that the *P. peruviana* nano-extract has greater efficacy in combating carbon tetrachloride (CCl_4_)-induced hepatotoxicity compared to the other extracts in rats. For example, it improved the antioxidant status of the lever by reducing elevated biochemical measurements, decreasing markers of the inflammatory response, and increasing the activity of antioxidant enzymes (catalase, glutathione peroxidase, and superoxide dismutase). Moreover, when taken orally, the *P. peruviana* nano-extract has been found to be safer than the other extracts. However, authors have not described which extracts have been used as positive control.

Withanolides, including physaperuvin G, physaperuvins I, and J, and others, isolated from *P. peruviana* fruits, have anti-inflammatory properties. They display the nuclear factor kappa-light-chain-enhancer of activated B cell (NF-κB) activity with stably transfected NF-κB Luc-293 human embryonic kidney cells induced by tumor necrosis factor-α (TNF-α). In addition, these compounds possess nitric oxide (NO) inhibitory activity against lipopolisyccharide (LPS)-stimulated NO release, and antiproliferative properties (in the HT-29 human colorectal cancer cell line model). Moreover, cytotoxicity has not been observed at the concentration of 50 µM [50]. However, more research is needed to determine the chemical compounds with anti-inflammatory potential isolated from *P. peruviana* fruits.

### 2.6. Antihepatotoxic, Antinephrotoxicity, and Other Activities (In Vivo)

Extracts from *P. peruviana* fruits have shown antihepatotoxic and antinephrotoxicity action in various models [51,52]. For example, Dkhil et al. [51] observed that the administration of a methanolic extract from *P. peruviana* fruits (200 mg/kg/day; for 5 days) reduces hepato-renal toxicity in rats treated with cadmium chloride (CdCl_2_) by reducing oxidative stress (lipid peroxidation) and improving the level of glutathione and the activity of enzymes in the kidney and liver. In addition, the used extract reversed histopathological changes in the liver and kidney, and also increased the expression of the Bcl-2. Taj et al. [52] analyzed the antihepatotoxic properties of *P. peruviana* whole ripe fruits and two extracts (ethanol and water) from *P. peruviana* fruits in normal as well as in carbon tetrachloride (CCl_4_)-intoxicated rats. In this experiment, rats were divided into five groups: (1) control rats (orally given distilled water—2 mL/kg b.w./day, for six days); (2) rats that were given standard drug Liv-52 (500 mg/kg b.w.); (3) rats that were orally administered with water extract (250 mg/kg b.w.); (4) rats that were orally administered with ethanol extract (250 mg/kg b.w.); and (5) rats that were orally administered with whole ripe fruits (23 g/kg b.w.). Authors observed that the water extract of *P. peruviana* had the highest antihepatotoxicity activity in both rat models while ripe fruit and ethanol extract demonstrated moderate activity compared to the standard drug. Arun and Asha [53] also noted the antihepatotoxic action of water, ethanol, and hexane extracts of *P. peruviana* (500 mg/kg b.w./day) against CCl_4_-induced hepatotoxicity in rats. However, the ethanol and hexane extracts demonstrated moderate activity compared to the water extract, which showed activity at a low dose of 125 mg/kg b.w./day. Histopathological changes induced by CCl_4_ were also significantly reduced by tested extracts. According to Abdel Moneim [54], the preventive effect of *P. peruviana* fruit juice against the toxicity of CCl_4_ on the reproduction system in rats was also noted. This juice supplementation significantly increased the testicular glutathione and significantly decreased the level of lipid peroxidation and the nitric oxide production compared with the CCl_4_ group. Moreover, the decline in the activity of antioxidant enzymes after CCl_4_ was ameliorated by *P. peruviana* fruit juice. Other authors found that *P. peruviana* fruit juice (5 and 15%) suppresses high-cholesterol-diet-induced hypercholesterolemia in rats [55]. Tested juices (after 2 months of supplementation) showed lower levels of total triacylglyceride, total cholesterol, and LDL cholesterol, as well as higher levels of HDL compared to animals fed with a high-cholesterol diet and cholesterol-free diet.

Recently, the protective action of two sucrose esters (peruviose A and B—the major metabolites of *P. peruviana* cape; 5, 10, 20 mg/kg/day, for 2 days) in 2,4,6-trinitrobenzene sulfonic acid (TNBS)-induced colitis model in rats was observed by Ocampo et al. [56]. Another study indicates that the extract of calyces that envelop *P. peruviana* fruits led to the increased absorption, distribution, and elimination of rutin as well as increased bioavailability of its metabolites in rabbits [57].

The main biological activities of *P. peruviana* fruits and their juice are presented in Figure 3.

## 3. Bioactive Compounds of Seeds and Their Biological Activity

The goldenberry fruit contains about 100–300 small seeds, which are distributed centrally and peripherally. The mean weight of these seeds is about 0.26 g. It is important that they have high values of carbohydrates (325.1 g/kg) and fiber (315.g/kg). In addition, goldenberry seeds are a good source of fatty acids, especially unsaturated fatty acids (about 90%), including linoleic acid, which has cardioprotective potential. Various phytosterols such as stigmasterol, campesterol, β-sitosterol, and others with cardioprotection action are also found in goldenberry seed oil [2,7,58].

Moreover, goldenberry seeds contain different phenolic compounds, which may determine the antioxidant activity of goldenberry seeds. They contain 2.3 and 20-fold higher than for goldenberry fruits and roots, respectively. For example, the total flavonoid concentration of the extract from these seeds is about 0.74 g/kg, which is 74% higher than for fruits [2,7,59,60].

The oil of goldenberry seed is also a source of four isoforms of tocopherol (α, β, ɤ, and δ), which have demonstrated to be beneficial to health, including antioxidant potential [2,7].

There are only a papers about the biological activity of *P. peruviana* seeds. For example, Erturk et al. [60] studied the antimicrobial potential of *P. peruviana* extracts with fruits, seeds, roots, and leaves. They observed that an extract from *P. peruviana* seeds has the most effective antimicrobial activity among all tested extracts. However, the seed extract had a moderate antimicrobial activity when compared with control drugs, such as nystatin, cephazolin, and ampicillin. It is important that this report did not describe the concentration of the used extracts.

Sharmila et al. [61] reported that dried *P. peruviana* seeds may be used in the treatment of glaucoma and jaundice. However, this information is only based on the testimonials of the local population in India.

## 4. Bioactive Compounds of Pomace and Its Biological Activity

Pomace (seeds and skins) is the part which is received after *P. peruviana* juice extraction (ca. 27% of fruit weight) [7]. The pomace contains 19.3% oil, 17.8% protein, 3.10% ash, 28.7% crude fiber, and 24.5% carbohydrates. It has a similar content of protein and lipid, including fatty acids, as *P. peruviana* seeds, but its protein content is higher by about 270% compared to pomace from pineapple [62]. On the other hand, the dietary fiber content of *P. peruviana* pomace is lower by about 50% than in *P. peruviana* seeds [7]. *P. peruviana* pomace is also the source of phytosterols, especially campesterol, and tocopherols. β-tocopherol was the main tocopherol [7].

The results of Ramadan [63] demonstrated that *P. peruviana* pomace reduces high-cholesterol-diet-stimulated hypercholesterolemia in male albino rats (N = 20). These results show a reduction in the levels of total cholesterol, total triacylglycerol, LDL-cholesterol, and an increase in the level of HDL-cholesterol. In this model, animals were supplemented with *P. peruviana* pomace at a dose of 100 g/kg. This author suggests that *P. peruviana* pomace goldenberry supplementation seems to protect the liver in response to oxidative stress as well as decrease the magnitude of fatty liver development in response to a high-cholesterol diet.

## 5. Toxicity and Adverse Effects of *P. peruviana* Fruits and Their Food Products

It is important that no papers have been found regarding the toxic effects of consumption of *P. peruviana* fruits, seeds, pomace powder, and its oil extract. However, Kasali et al. [4] indicate that *P. peruviana* extracts have weak toxicity (LD_50_ > 500 mg/kg). For example, Perk et al. [64] studied the acute and subchronic toxic action of *P. peruviana* fruit juice (oral dose of 0.1, 1 and 5 g/kg/day (in a volume of 1 mL/100 g); for 14–90 days) in Wistar rats. They observed that the tested juice does not induce renal, hepatic, hematological toxicity, and genotoxicity, but cardiotoxicity was only observed in male Wistar rats—the potassium level was significantly increased in the male group treated with 5 g/kg of lyophilized fruit juice. In addition, the plasma levels of troponin I and troponin T increased significantly.

Recently, Acar [65] studied the protective action of *P. peruviana* fruit extract against the toxic effects of sodium salt of glutamic acid and monosodium glutamate, which are used as additives to improve flavor in food products. The genetic and biochemical effects of tested food additives, antimutagenic activity, and the protective role of *P. peruviana* fruit extract (125 and 250 mg/L) against these effects were investigated with *Allium cepa* L. test material. The author observed that the used extract had concentration-dependent therapeutic effects in lowering the toxicity of the food additives. For example, the used plant extract decreased oxidative stress (including decreasing lipid peroxidation and increasing GSH level) and DNA damage (based on the formation of the DNA tail by the comet test), and this also manifested itself in physiological parameters.

## 6. Conclusions

For the first time, this mini-review paper indicates that *P. peruviana* fruits and their various food products, especially juice, may be good candidates as functional foods with beneficial properties on chronic conditions, such as diabetes and others. This paper also demonstrates various biological activities of *P. peruviana* fruits, their food products, and the pure chemical compounds extracted from them both in vitro and in vivo. For example, the supplementation of *P. peruviana* fruits has been reported to decrease oxidative stress, attenuate inflammation, reduce blood glucose level, and others. However, the number of these models is too limited to unequivocally indicate that *P. peruviana* fruits and their food products have beneficial action in humans. Moreover, the pro-healthy properties of commercial food products have not been well described in scientific literature. There are only a few papers about *P. peruviana* fruit juice. Therefore, more randomized clinical trials with larger groups are needed—especially both healthy people and those with various diseases—on the treatment or prophylaxis of diabetes, cancer, cardiovascular diseases, and others. Such studies should also examine their long-term actions and safety.

In addition, various studies employed different types of *P. peruviana* fruit preparation, especially extracts, the composition of which was not always determined, and their bioactive ingredients also were, in many cases, not clearly identified. On the other hand, high phytochemical contents, especially phenolic compounds, sucrose esters, withaperuvins, and phenylpropanes appear to determine the main biological properties of *P. peruviana* fruits. Therefore, different preparations from *P. peruviana* fruits as valuable sources of bioactive phytochemicals could be used as functional foods, but the process of introducing these fruits in supplements or conventional medicine must be preceded in various in vivo models. Moreover, there is a need for more studies that would clarify the exact mechanisms of action and determine which bioactive compounds are responsible for the beneficial action of *P. peruviana* fruits and their food products.

## Figures and Tables

**Figure 1 ijms-26-03493-f001:**
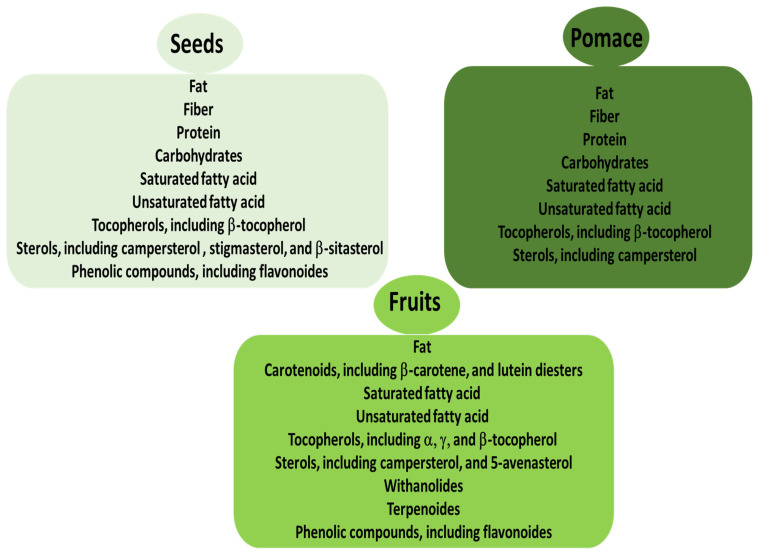
Selected bioactive compounds of *P. peruviana* fruits, seeds, and pomace.

**Figure 2 ijms-26-03493-f002:**
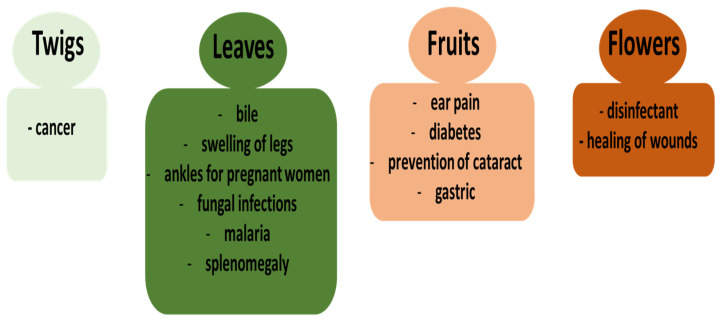
Uses of various parts of *P. peruviana* in ethnomedicine.

**Figure 3 ijms-26-03493-f003:**
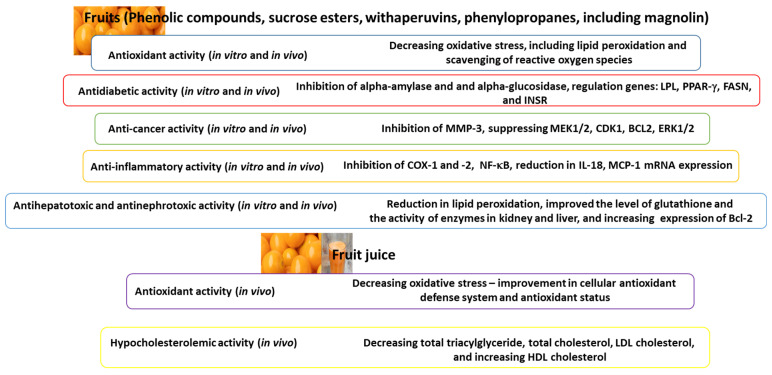
Main biological activities of *P. peruviana* fruits and their juice. BCL2—B-cell lymphoma 2; CDK1—cyclin-dependent kinase 1; COX—cyclooxygenase; ERK1/2—mitogen-activated protein (MAP) kinase; FASN—fatty acid synthase; HDL—high-density lipoprotein; Il-18—interleukin-18; INSR—insulin receptor; LDL—low-density lipoprotein; LPL—lipoprotein lipase; MCP-1—monocyte chemo-attractant protein; MEK1/2—MAP/ERK kinase; MMP-3—matrix metalloproteinase-3; NF-κB—kappa-light-chain-enhancer of activated B cells; PPAR-γ—peroxisome proliferator-activated receptor gamma.

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
