# Peer review of "Physalis peruwiana Fruits and Their Food Products as New Important Components of Functional Foods"

_ijms, 2025, doi:10.3390/ijms26083493_

Round 1

Reviewer 1 Report

Comments and Suggestions for Authors

Please reformulate the paragraph between lines 31-38. Besides the fact that the percentage exceeds 100%, it refers to different things, namely the carbohydrate, protein and fat content of the waste, but the vitamin E content of the juice.

Maybe you should check the values ​​for unsaturated fatty acids, which are almost 4 times more than the total fat content, also correct in figure 1, pomace not "pamace"

Why rows 68-72 are written differently? The author should detail in her article the utilization of P. peruviana  as  ethnotherapeutic plants, without  sending readers to read another article.

Row 88: what is polame?

reformulate the paragraph from line 140 to 149 it is very ambiguous. Who is the salt stress on (mice or plant)? What happens to the glucose level? is not clearly formulated.

Please rephrase the sentence from lines 176-178, it is difficult to understand.

please be a little more precise, it is not clear in the sentence (287-289) whether those are the concentrations of the extracts or the doses administered. Rephrase

Author Response

Thank you for reviewing the manuscript and providing such helpful comments. All of them have been taken into consideration when revising the manuscript.

Please reformulate the paragraph between lines 31-38. Besides the fact that the percentage exceeds 100%, it refers to different things, namely the carbohydrate, protein and fat content of the waste, but the vitamin E content of the juice.

Response: I have corrected it. Now, it is: “Mokhar et al. (2018) indicate the following composition of P. peruviana powder waste: 16.7% dietary fiber, 61% carbohydrates, 15.9% protein, 13.7% fat, 5.9% moisture, and 3.5% ash.”

I have added information about the vitamin E content of fruits in the chapter - Bioactive compounds of fruits and their biological activity: “The fruits contain appreciable amount of lipids and carotenoids (especially β-carotene), with some tocopherols (about 17 mg/100 g of fruit FW, mainly ɤ-, α-, and β-tocopherol) phytosterols (about 10 mg/100 g of fruit FW, mainly campesterol and 5-avenasterol), and lutein diesters (Breithaupt et al., 2002; Ramadan et al., 2003; Etzbach et al., 2018).”

Maybe you should check the values ​​for unsaturated fatty acids, which are almost 4 times more than the total fat content, also correct in figure 1, pomace not "pamace"

Response: I have corrected and modified figure 1. Now, it is “pomace”. I have also removed the values of bioactive compounds. However, Nocetti, D., Nunez, H., Puente, L., Espinosa, A., Romero, F. (“Composition and biological effects of goldenberry byproducts: an overview”. J Sci Food Agric 2020;100:4335-4346.) described the same values.

Why rows 68-72 are written differently?

Response: I have corrected.

The author should detail in her article the utilization of P. peruviana  as  ethnotherapeutic plants, without  sending readers to read another article.

Response: I have added more information about it. For example, “Various parts of this plant, including twigs, leaf, stem, fruits, flowers, roots, seeds, and other are often used in traditional medicine in different countries (for example, Cameroon, Colombia, India, Kenya, Nepal, Tanzania, Uganda, and other), but leaf is the mostly used part (49.3%), prepared by decoction (31.6%) and overall administrated orally (53.6%) as the main route of admission (Chang et al., 2016; Nocetti et al., 2020; Kasali et al., 2021; Navarro-Hoyos et al., 2022). Uses of various parts of P. peruviana in ethnomedicine are summarized in Figure 2. For example, they have  antispasmodic, antidiabetic, diuretic, antiseptic, sedative, and analgesic properties. In addition, they are also used to treat hepatitis, malaria, dermatitis, asthma, and rheumatism.

I have also presented this information on Figure 2 – “Uses of various parts of P. peruviana in ethnomedicine.”

Row 88: what is polame?

Response: It should be “pomace”, and I have corrected it.

reformulate the paragraph from line 140 to 149 it is very ambiguous. Who is the salt stress on (mice or plant)? What happens to the glucose level? is not clearly formulated.

Response: I have added more information about it: “Recently, Gonzalez-Buenrostro et al. (2024) have studied the effect of saline stress (10 – 40 mM) on the metabolic profile and antidiabetic activity of P. peruviana fruits. They observed that saline stress (especially 40 mM NaCl) increases the phenolic content in P. peruviana fruits (for example, about 39% increase for the 40 mM NaCl treatment). On the other hand, fruits (treated with saline stress during cultivation) did not significantly decrease the hyperglycaemic peak in healthy rats.”

Please rephrase the sentence from lines 176-178, it is difficult to understand.

Response: I have corrected this sentence: “It is important that magnolin had much less cytotoxicity in comparison with doxorubicin (for the concentration 5 µg/mL: 6.96% growth inhibition (for magnolin), and 30.48% growth inhibition (for doxorubicin)) towards dermal fibroblasts (in vitro). In this experiment, MTT (3-(4, 5-dimethylthiazolyl-2)-2, 5-diphenyltetrazolium bromide) assay was used.”

please be a little more precise, it is not clear in the sentence (287-289) whether those are the concentrations of the extracts or the doses administered. Rephrase

Response: I have corrected this sentence and added more information: “Taj et al. (2014) analyzed the antihepatotoxic properties of P. peruviana whole ripe fruits and two extracts (ethanol and water) from P. peruviana fruits in normal as well as in carbon tetrachloride (CCl4) intoxicated rats. In this experiment, rats were divided into five groups: (1) control rats (orally given distilled water – 2 mL/kg b.w./day, for six days); (2) rats were given standard drug Liv-52 (500 mg/kg b.w.); (3) rats were orally administered with water extract (250 mg/kg b.w.); (4) rats were orally administered with ethanol extract (250 mg/kg b.w.); and (5) rats were orally administered with whole ripe fruits (23 g/kg b.w.). Authors observed that the water extract of P. peruviana has the highest antihepatotoxicity activity in both rat models while ripe fruit and ethanol extract demonstrate moderate activity compared to standard drug.”

Reviewer 2 Report

Comments and Suggestions for Authors

I was invited to review the paper Physalis peruwiana fruits and their food products as new important components of functional foods, which is part of the review group. The paper was edited and delivered in PDF format, so I will write a summarised review. At first glance, the paper is satisfactorily written and only needs a little aesthetic refinement (e.g. lines 68-72 are in the wrong font, double spacing between words,...). One soon gets the impression that the work is a compilation of previous works, lacking a critical conclusion by the author, so I would not recommend the work for publication in this form. The only scientific method used by the author is the search for more recent scientific literature, so in this case a systematisation and review of the same is required. The data on the bioactive components (BC) of the plant P. peruviana are listed in order, which seems confusing as they refer to different parts of the plant, perhaps a tabulation of the main BCs together with the method of isolation and type of quantification would be useful here. The antioxidant and antitumor activity is a string of enumerations of experiments performed so far, with the chemical components that are carriers of this activity mostly not specified (mostly extracts, we don't even know which solvent is involved). Things look a little better for the antidiabetic and anti-inflammatory effects, but here too there is no linking of the results, which I would expect in a review paper, as that is its only main task. The conclusion, like the whole article, is general and can be applied to any other paper.

Author Response

I was invited to review the paper Physalis peruwiana fruits and their food products as new important components of functional foods, which is part of the review group. The paper was edited and delivered in PDF format, so I will write a summarised review. At first glance, the paper is satisfactorily written and only needs a little aesthetic refinement (e.g. lines 68-72 are in the wrong font, double spacing between words,...).

Response: Thank you for reviewing the manuscript and providing such helpful comments. All of them have been taken into consideration when revising the manuscript.

I have corrected it.

One soon gets the impression that the work is a compilation of previous works, lacking a critical conclusion by the author, so I would not recommend the work for publication in this form. The only scientific method used by the author is the search for more recent scientific literature, so in this case a systematisation and review of the same is required.

Response: Few review papers have explored the ethnotherapeutic uses of various parts of P. peruviana (Nocetti et al., 2020; Kasali et al., 2021), but these papers generally do not include P. peruviana fruits and their food products as an important components of functional foods. For example, a review by Nocetti et al. (2020) only describes the biological activities of selected goldenberry byproducts, including seeds and pomace, which is produces after juice extraction. Therefore, the aim of the present mini-review is to provide an overview of the beneficial potential of P. peruviana fruits and their food products as an important components of functional foods.

The data on the bioactive components (BC) of the plant P. peruviana are listed in order, which seems confusing as they refer to different parts of the plant, perhaps a tabulation of the main BCs together with the method of isolation and type of quantification would be useful here.

Response: I have described the bioactive components of P. peruviana fruits, seeds, and pomace in various chapters:

“2.1. Bioactive compounds of fruits and their biological activity

The fruits contain appreciable amount of lipids and carotenoids (especially β-carotene), with some tocopherols (about 17 mg/100 g of fruit FW, mainly ɤ-, α-, and β-tocopherol) phytosterols (about 10 mg/100 g of fruit FW, mainly campesterol and 5-avenasterol), and lutein diesters (Breithaupt et al., 2002; Ramadan et al., 2003; Etzbach et al., 2018).

Various withanolides and triterpenes were not only isolated from leaves, roots, but also from P. peruviana fruits, but at low concentration (Llano et al., 2017; Aminah et al., 2021). In addition, the extract from P. peruviana fruits contained various flavonoids (including kaempferol, quercetin, rutin, and myricetin), sucrose esters (peruvioses and nicoandrose), and other chemical compounds such as blumenol A, hydroxyester disaccharides, carbohydrate ester of cinnamic acid, glycosically bound compounds, (+)-(S)-dehydrovomifoliol, and loliolide with different biological activity (Aminah et al., 2021). Moreover, P. peruviana fruits are the source of trans-resveratrol, which is even richer than red wine (Lotz and Spangenberg, 2016).”

“3. Bioactive compounds of seeds and their biological activity

The goldenberry fruit contains about 100 – 300 small seeds, which are distributed centrally and peripherally. The mean weight of these seeds is about 0.26 g. It is an important that they have high values of carbohydrates (325.1 g/kg) and fiber (315.g/kg). In addition, the goldenberry seeds are a good source of fatty acids, especially unsaturated fatty acids (about 90%), including linoleic acid, which has cardioprotective potential. Various phytosterols such as stigmasterol, campesterol, β-sitosterol, and other with cardioprotection action are also found in goldenberry seed oil (Kiran and Prajapati, 2017; Mokhtar et al., 2018; Nocetti et al., 2020).

Moreover, the goldenberry seeds contain different phenolic compounds, which may decide about antioxidant activity of goldenberry seeds. They contain 2.3 and 20-fold higher than for goldenberry fruits and roots, respectively. For example, the total flavonoid concentration of the extract from these seeds is about 0.74 g/kg, which is 74% higher than for fruits (Cardona et al., 2017; Erturk et al., 2017; Mokhtar et al., 2018; Nocetti et al., 2020).

The oil of goldenberry seed is also source of four isoforms of tocopherol (α, β, ɤ, and d), which have demonstrated to the beneficial to health, including antioxidant potential (Mokhtar et al., 2018; Nocetti et al., 2020).”

“4. Bioactive compounds of pomace and its biological activity

Pomace (seeds and skins) is the part, which is received after P. peruviana juice extraction (ca. 27% of fruit weight) (Nocetti et al., 2020). The pomace contains 19.3% oil, 17.8% protein, 3.10% ash, 28.7% crude fiber and 24.5% carbohydrates. It has similar content of protein and lipid, including fatty acids as P. peruviana seeds, but its protein content is higher by about 270%, compared to pomace from pineapple (Nagarajaiah and Prakash, 2016). On the other hand, the dietary fiber content of P. peruviana pomace is lower by about 50% than in P. peruviana seeds (Nocetti et al., 2020). P. peruviana pomace is also the source of phytosterols, especially campesterol, and tocopherols. b-tocopherol was the main tocopherol (Nocetti et al., 2020).”

In addition, I have also presented the selected bioactive compounds of  P. peruviana  fruits, seeds and pomace on Figure 1.

I have also added more information about phytochemical methods in the chapter of Introduction. For example, “AOAC standard methods have been very often used for phytochemical analysis of P. peruviana fruits (for fat, protein carbohydrate, fiber, and other) (Vaillant et al., 2021).”; For example, vitamin C was assessed by high-performance liquid chromatography (HPLC) method by Vaillant et al. (2021).”; “However, the health benefits of this plant are related to the content of various bioactive chemical compounds, including 76 withanolides, 5 phenolic compounds - flavonoids, 2 alkaloids, 14 sucrose ester, and other such as vitamins, especially carotenoids, and physalins, which were identified by chromatographic methods (Aminah et al., 2021).”; Standard techniques have been used for phytochemical analysis of P. peruviana fruits or other plant parts, including high-performance liquid chromatography (HPLC) with ultraviolet (UV) detection, liquid chromatography-tandem mass spectrometry (LC-MS/MS, gas chromatography-mass spectrometry (GC-MS), and liquid chromatography-mass spectrometry (LC-MS) (Vaillant et al., 2021). “

The antioxidant and antitumor activity is a string of enumerations of experiments performed so far, with the chemical components that are carriers of this activity mostly not specified (mostly extracts, we don't even know which solvent is involved). Things look a little better for the antidiabetic and anti-inflammatory effects, but here too there is no linking of the results, which I would expect in a review paper, as that is its only main task.

Response: I have modified these chapters. For example, I have added more information about solvent, but authors often do not described the phytochemical characteristic of used plant extract and used solvents.

I have also added information that “…the compounds responsible for anti-cancer properties are not well identified due to the considerable variation present in the chemical composition of P. peruviana fruit extracts, which can depend on inter alia time of harvest and extraction technique. Only few papers have described these properties. Sayed et al. (2022) suggest that magnolin (isolated from 95% ethanol extract of P. peruviana fruits) is a potent preferential antiproliferative compound against the human pancreatic cancer cell line PANC-1 with and IC50 of 0.51 ± 0.46 µM (it was comparable with positive control – doxorubicin (IC50 of 0.17 ± 0.15 µM)…..”.

The conclusion, like the whole article, is general and can be applied to any other paper.

Response: I have modified the chapter of Conclusion: “ For the first time, this mini-review paper indicate that P. peruviana fruits and their various food products, especially juice may be good candidates as functional foods with beneficial properties on chronic conditions, such as diabetes and other. This paper also demonstrates various biological activities of P. peruviana fruits, their food products, and the pure chemical compounds extracted from them both in vitro and in vivo. For example, the supplementation of P. peruviana fruits has been reported to decrease oxidative stress, attenuate inflammation, reduce blood glucose level, and other. However, the number of these models is too limited to unequivocally indicate that P. peruviana fruits and their food products have beneficial action in human. Moreover, the pro-healthy properties of commercial food products have not been well described in scientific literature. There are only few papers about P. peruviana fruit juice. Therefore, more randomized clinical trials with larger groups are needed, especially both healthy people and those with various diseases – on the treatment or prophylaxis of diabetes, cancer, cardiovascular diseases, and other. Such studies should also examine their long-term actions and safety.

In addition, various studies employed different types of P. peruviana fruit preparation, especially extracts, the composition of which was not always determined, and their bioactive ingredients also were, in many cases, not clearly identifies. On the other hand, high phytochemical contents, especially phenolic compounds, sucrose esters, withaperuvins, and phenylpropanes appear to decide about main biological properties of P. peruviana fruits. Therefore, different preparations from P. peruviana fruits as valuable sources of bioactive phytochemicals could be used as functional foods, but the process of introducing of these fruits in supplements or conventional medicine must be preceded in various in vivo models. Moreover, there is a need for more studies that would clarity the exact mechanisms of action and determine which bioactive compounds are responsible for beneficial action of P. peruviana fruits and their food products.

Round 2

Reviewer 2 Report

Comments and Suggestions for Authors

Dear Autor,
I propose accepting the work in this form.